# Antenatal and perinatal service delivery associations with breastfeeding outcomes in Nepal: Analysis of the 2016 Nepal Demographic and Health Survey

**Breanna Louise Hollow[1], Max K. Bulsara[2], Prakash Dev Pant[3], Hilary Jane Wallace**[1,4]*

**1** School of Medicine, The University of Notre Dame Australia, Fremantle, Australia, **2** Institute for Health Research, The University of Notre Dame Australia, Fremantle, Australia, **3** Monitoring and Evaluation Consultant, Kathmandu, Nepal, **4** School of Population and Global Health, The University of Western Australia, Crawley, Australia

* hilary.wallace@uwa.edu.au

**Data Availability Statement:** Nepal Demographic and Health Survey 2016 [Dataset]. NPKR7HFL.SAV (kids recode); NPIR7HFL.SAV (individual recode). Rockville, Maryland: Ministry of Health, New ERA,

## Abstract

Infant and Young Child Feeding (IYCF) breastfeeding guidelines of the World Health Organization (WHO) have been promoted in Nepal since the early 1990s. This study investigated whether antenatal and perinatal service delivery in Nepal are associated with early initiation of breastfeeding and age-appropriate feeding practice (exclusive breastfeeding to six months; introduction of complementary foods at six months with continued breastfeeding to two years). Data from the 2016 Nepal Demographic and Health Survey (NDHS) were analysed using multivariable logistic regression. The unit of analysis was an interviewed woman and her last-born child aged 0–23 months. We examined number of antenatal visits, place and type of delivery, infant-mother skin-to-skin contact post-delivery, and breastfeeding observation and counselling by a healthcare provider within two days post-delivery. Of 1938 mother-infant dyads, 1073 (55.4%) commenced breastfeeding within one hour of delivery and 1665 (85.9%) were engaged in age-appropriate feeding. Breastfeeding within one hour of delivery was associated with infants delivered vaginally (aOR: 4.76, 95% CI: 2.96–7.65), infant-mother skin-to-skin contact post-delivery (aOR:2.10, 95% CI: 1.63–2.72) and observation of breastfeeding by a healthcare provider within two days post-delivery (aOR: 1.58, 95% CI: 1.20–2.08). Age-appropriate feeding was lowest amongst mothers with infants aged 4–5 months (40.8%) compared to those with infants aged 0–1 month (aOR: 0.158, 95% CI: 0.083–0.302). Antenatal and perinatal service delivery were not significantly associated with age-appropriate feeding. Further promotion of infant-mother skin-to-skin contact post-delivery (including after caesarean delivery) and observation of early breastfeeding may increase the rate of breastfeeding within one hour of delivery. Promotion of exclusive breastfeeding in antenatal and perinatal services and **a**dditional postnatal support should be considered to increase exclusive breastfeeding of infants to six months. These improvements may be achieved through enhanced implementation of the Baby Friendly Hospitals Initiative and effective training and sufficient practice for skilled birth attendants.

and ICF [Producers]. ICF [Distributor], 2017. Available from https://dhsprogram.com/data/dataset/Nepal_Standard-DHS_2016.cfm?flag=0.

**Funding:** The authors received no specific funding for this work.

**Competing interests:** The authors have declared that no competing interests exist.

## Introduction

In 2003 the World Health Organisation (WHO), in collaboration with UNICEF, developed the Infant and Young Child Feeding (IYCF) guidelines [1] based on research underscoring the foundational role of breastfeeding in improving child development, mortality and morbidity [2]. These guidelines recommend that all mothers should be supported to:

1. Initiate breastfeeding as soon as possible after birth, ideally, within the first hour after delivery;

2. Exclusively breastfeed their infants for the first six months of life, and;

3. Introduce nutritionally adequate complementary (solid) foods at six months, together with continued breastfeeding up to two years of age or beyond.

Systematic reviews have consistently shown that breastfeeding within one hour of delivery is linked with lower all-cause and infection-related neonatal mortality [3, 4]. In a multinational meta-analysis of 136,047 breastfed newborns [5], newborns breastfed within the first hour post-delivery had a 33% lower risk of dying than those breastfed between two and 23 hours after birth. Exclusive breastfeeding up to six months is associated with a lower incidence of infantile gastrointestinal infection [6], the second leading cause of post-neonatal deaths in children under five [7]. Similarly, exclusive breastfeeding to six months with breast and complementary feeding to 24 months, has been linked to lower rates of malnutrition and childhood stunting [8].

Nepal adopted several public health policies over the past three decades in an attempt at increasing breastfeeding practice according to IYCF recommendations. In 1992, the *Mother's Milk Substitutes Act*, which established a Breastfeeding Protection and Promotion Committee (BPPC), saw health service collusion with milk-substitute manufacturers criminalised, and formalised the place of breastfeeding promotion strategies within hospitals [9]. Two years later the UNICEF Baby Friendly Hospitals Initiative (BFHI) [10] was implemented and increased breastfeeding coverage across South Asia [11], a previously low-uptake region [12]. The program's success is primarily attributed to its birth attendant training course, coupled with the facilitation of infant-mother skin-to-skin contact post-delivery [11, 13]. A core skill in the National Policy on Skilled Birth Attendants (Nepal) [14] is that a skilled birth attendant will "assist women and their newborns in initiating and establishing early and exclusive breastfeeding, including educating women and their families and other helpers in maintaining successful breastfeeding". In 2006 the rate of breastfeeding within one hour of delivery was 35% and the percentage of infants who were breastfed exclusively up to six months, 53% [15]. Some improvement was observed in 2011, where the figures neared 45% and 70% respectively [16].

Previous studies have explored demographic factors contributing to breastfeeding practice in Nepal, but few have examined the association with exposure to antenatal and perinatal service delivery [12, 17–19]. Demographic and Health Survey (DHS) data is a powerful population-based resource to examine health service delivery and the association with maternal and child health outcomes. This study aimed to evaluate whether specific elements of antenatal and perinatal service delivery (number of antenatal visits, type and place of delivery, infant-mother skin-to-skin contact post-delivery, and breastfeeding counselling and observation within the first two days post-delivery) are positively associated with the breastfeeding within one hour of delivery and current coverage with age-appropriate feeding practice in Nepali infants up to 24 months of age.

## Methods

### Data sources

Data were obtained from the 2016 Nepal Demographic and Health Survey (NDHS), a nationally representative cross-sectional household survey, funded by the US Agency for International Development (USAID) [20]. The 2016 NDHS sampling frame was a modified version of the Nepal Central Bureau of Statistic's 2011 National Population and Housing Census [20]. Data were stratified and selected in two stages in rural areas and three stages in urban areas. In urban areas, wards were the primary sampling units (PSU), from which one enumeration area (EA) was derived. Households were subsequently selected from EAs. In rural areas, wards were the PSU from which households were selected directly. Only households containing a woman aged 15–49 the night before survey administration were eligible for interview. Of the 13089 women eligible, 12862 completed the survey, representing a response rate of 98% [20]. For results to be nationally, regionally, and provincially representative, weighting factors have been calculated and added to the DHS datafiles. Full details of the NDHS sampling design are discussed elsewhere [20].

The present study used the Children's Recode DHS file (KR file) with the addition of 11 food security variables from the Individual Women's Data file (IR file). The KR file includes one record for every child (alive or dead) of an interviewed woman born in the five years preceding the survey [20]. In this study, only last-born children up to 23 completed months, who were still alive and living with their mother, were included for analysis (n = 1930).

### Outcome variables

The outcome, early initiation of breastfeeding, defined by WHO as "breastfeeding within the first hour of life" [21], was derived from the DHS question; "How long after birth did you first put (name of child) to the breast?" [20]. Infants put to the breast within one hour of birth were deemed to comply with early initiation of breastfeeding.

The outcome, adherence to IYCF-recommended feeding practice, was defined according to the child's age. For children 0–5 completed months of age (i.e., under six months), feeding practice was appropriate if the child was exclusively breastfed at the time of survey administration. A child was determined to be "exclusively breastfed" if they had received breastmilk in the 24 hours preceding the interview but nothing else. Feeding with non-breastmilk liquids (including water) in this period, precluded children from this group [22]. For children aged 6–23 completed months (i.e., 6 months to under 24 months), feeding practice was deemed appropriate if the child had received breastmilk and "complementary foods" in the 24 hours prior to survey administration. According to WHO guidelines, complementary foods include both solid and semi-solid foodstuffs [1].

### Explanatory variables

Explanatory variables for breastfeeding outcomes were adapted from the conceptual framework of Bhandari et al. [18] (see Fig 1).

Variables related to antenatal and perinatal service delivery were informed by the WHO breastfeeding practice guidelines [21] and the WHO recommendations on antenatal care for a positive pregnancy experience [23]: number of antenatal visits (0 visits; 1–3 visits; 4 visits; 5–7 visits; 8+ visits), place of delivery (health facility; not in a health facility), type of delivery (vaginal; caesarean), infant-mother skin-to-skin contact post-delivery (child put directly on the bare skin of the mother's chest immediately after the birth) (yes; no), breastfeeding counselling (yes; no) by any healthcare provider within the first two days post-delivery, and breastfeeding observation (yes; no) by any healthcare provider within the first two days post-delivery.

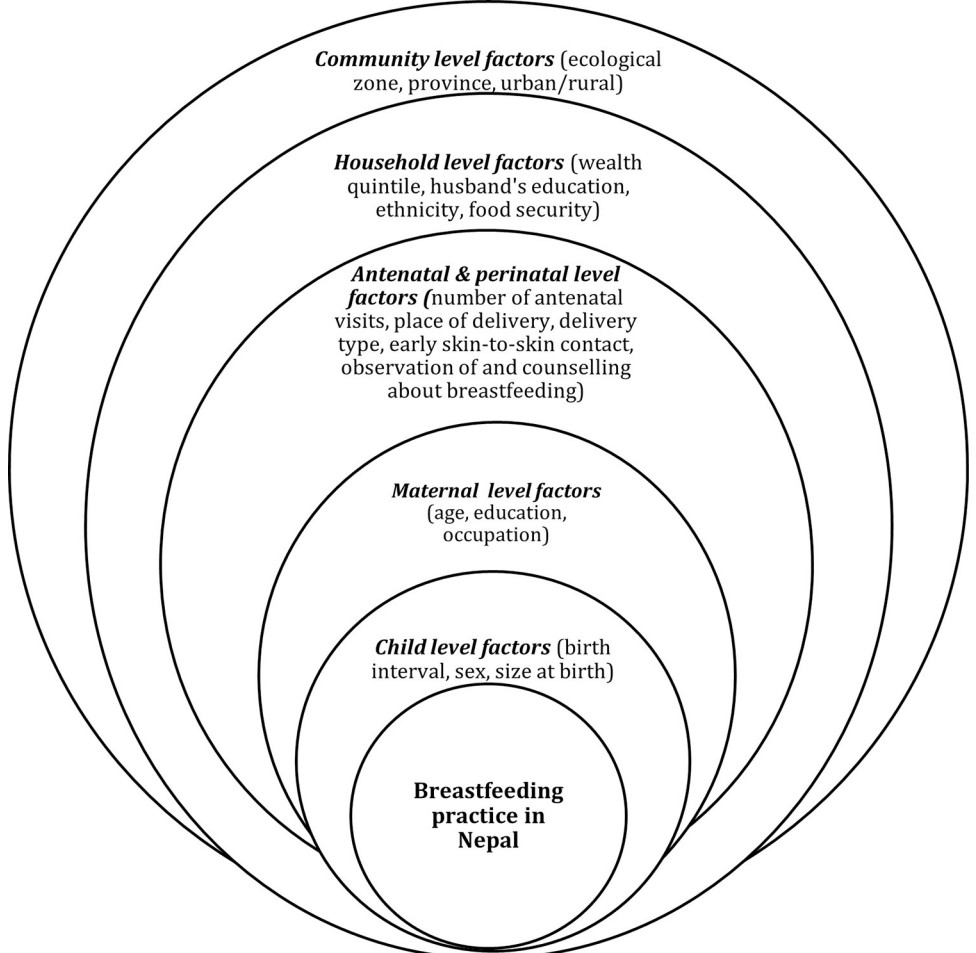

**Fig 1. Adapted conceptual framework for breastfeeding outcomes in Nepal [18].**

The selection of potential confounding variables was informed by literature review and identification of factors shown to be associated with breastfeeding practice in Nepal previously [12, 17–19]. Child-level factors included sex (m; f), age, size at birth (very small/small; average; large) and birth interval (first birth episode; 18 months or less; 19–36 months; 37 months+). Maternal and perinatal level factors included mother's age (15–19; 20–29; 30–39; 40–49), mother's occupation (did not work/household duties; agricultural work [paid and unpaid]; non-agricultural work [paid]), mother's highest education level (no education; primary; secondary; higher education), and place of delivery (home; health facility). Household level factors included the wealth quintile (poorest; poorer; middle; richer; richest), husband's highest education level (no education; primary; secondary; higher education), ethnicity (Brahman/Chhetri; other Terai castes; Dalits; Newar; Janajati; Muslim and other) and food security (secure; not secure). The food security variable was coded in accordance with Pandey & Fusaro [24]. Food was "secure" if the summed Household Food Insecurity Access Scale (HFIAS) score was 0; food was "insecure" for any score between 1 and 27 [24]. Community level factors included ecological zone (mountains; hills; Terai), province (1–7) and place of residence (rural; urban).

## Statistical analysis

Statistical analyses were performed using IBM SPSS Statistics for Windows, version 26 (IBM Corp., Armonk, N.Y., USA). Data were weighted using sampling weights in accordance with DHS guidelines [25]. All analyses used the Complex Sample Analysis module to account for the multi-stage sample design [25].

All variables were treated as categorical variables. Descriptive summaries were displayed using frequencies and percentages. Cross-tabulations and Chi-square tests were performed for the two binary outcome variables, breastfeeding within one hour of delivery and age-appropriate feeding practice, by each explanatory variable. Explanatory variables with a p-value of < 0.1 were included in univariable and multivariable logistic regression analyses. The child's sex was included a priori in the regression analyses [18]. Unadjusted and adjusted odds ratios (OR) were computed with 95% confidence intervals. In multivariable model testing, potential collinearity of explanatory variables was explored using Chi-square tests when the coefficient estimates were very unstable. Where two variables were highly associated with each other, one of the variables was removed from the multivariable logistic regression model.

Since the recommendations for infant and young child feeding change at 6 months of age, we subsequently applied the same analytical approach to two subgroups based on age: 0–5 completed months (exclusive breastfeeding) and 6–23 completed months (continued breastfeeding with complementary foods).

## Ethical approval

The 2016 NDHS survey protocol was approved by the Nepal Health Research Council (NHRC) and the ICF Institutional Review Board (ICF-IRB). The present study received ethics approval from the University of Notre Dame Australia, Fremantle, Human Research Ethics Committee (Ref. 2020-065F). Permission for the use of the 2016 NDHS dataset was granted by the DHS Program. Prior to the DHS interview, participants provided informed consent [22].

## Results

Table 1 presents the socio-demographic characteristics of the sample population. Of the 1938 children included in this study, 1043 (53.8%) were male and 895 (46.2%) were female. Two-thirds of mothers were 20–29 years of age (n = 1311; 67.7%), with 284 (14.7%) aged 15–19 years. About half of mothers either did not work or performed household duties (n = 913; 47.1%), while 41.4% were engaged in agricultural work (n = 803) and 11.4% in non-agricultural paid work (n = 221). The sample contained a mixture of urban (53.8%) and rural (46.2%) residents.

More than half of the women delivered in health facilities (n = 1178; 60.8%) and most had vaginal deliveries (n = 1742; 89.9%). Infant-mother skin-to-skin contact post-delivery was reported to have occurred in only 63.0% (n = 1222) of births. Just over half (n = 1049; 54.1%) received some counselling regarding breastfeeding from a healthcare provider during the first two days post-delivery and 50.3% (n = 974) were observed breastfeeding by a healthcare provider during the first two days post-delivery. Nearly three quarters (71.5%) of women attended at least 4 antenatal checks (n = 1386), with only 8.9% attending the WHO-recommended [23] 8 or more visits (n = 172). Overall, 55.4% of infants were breastfed within one hour of delivery, but in infants born by caesarean delivery the rate was only 22.6%. The variables significantly associated with breastfeeding within one hour of delivery in univariable analysis included: mother's occupation (p = 0.011), place of delivery (p<0.001), type of delivery (p<0.001), infant-mother skin-to-skin contact post-delivery (p<0.001), counselling regarding breastfeeding within two days post-delivery by a healthcare provider (p = 0.002), observation of

**Table 1. Characteristics of the sample population and distribution of factors influencing breastfeeding within the first hour of birth and age-appropriate infant and young child feeding (IYCF) practice.**

| | | N (%) | Breastfeeding within first hour N (%)[b] | P-value[c] | Age-appropriate feeding practice N (%)[b] | P-value[c] |
|---|---|---|---|---|---|---|
| **Total** | | 1938 (100) | 1073 (55.4) | | 1665 (85.9) | |
| ***Child factors*** | | | | | | |
| *Sex*[d] | | | | 0.146 | | 0.978 |
| | Male | 1043 (53.8) | 560 (53.7) | | 895 (85.9) | |
| | Female | 895 (46.2) | 513 (57.3) | | 769 (85.9) | |
| *Current Age (months)* | | | | | | **<0.0005** |
| | 0-1month | 159 (8.2) | N/A | | 127 (79.4) | |
| | 2-3months | 160 (8.2) | | | 115 (72.3) | |
| | 4-5months | 124 (6.4) | | | 51 (40.8) | |
| | 6-8months | 235 (12.1) | | | 196 (83.4) | |
| | 9-11months | 264 (13.6) | | | 250 (95.1) | |
| | 12-17months | 504 (26.0) | | | 482 (95.6) | |
| | 18-23months | 492 (25.4) | | | 444 (90.2) | |
| *Size at birth*[d] | | | | 0.199 | | 0.627 |
| | Very small/small | 331 (17.1) | 172 (52.0) | | 290 (87.6) | |
| | Average | 1318 (68.0) | 727 (55.2) | | 1124 (85.3) | |
| | Large | 288 (14.9) | 173 (60.1) | | 250 (86.8) | |
| *Birth interval* | | | | 0.828 | | 0.158 |
| | First birth episode | 792 (40.9) | 437 (55.2) | | 684 (86.4) | |
| | 18 months or less | 100 (5.2) | 60 (60.0) | | 79 (79.0) | |
| | 19-36months | 428 (22.1) | 233 (54.3) | | 360 (84.1) | |
| | 37months+ | 617 (32.0) | 343 (55.6) | | 542 (87.7) | |
| ***Maternal factors*** | | | | | | |
| *Current age (years)*[d] | | | | 0.575 | | **0.016** |
| | 15–19 | 284 (14.7) | 157 (55.3) | | 229 (80.6) | |
| | 20–29 | 1311 (67.7) | 728 (55.5) | | 1136 (87.0) | |
| | 30–39 | 314 (16.2) | 176 (56.1) | | 271 (86.3) | |
| | 40–49 | 29 (1.5) | 12 (41.4) | | 28 (100)[b] | |
| *Occupation* | | | | **0.011** | | **0.001** |
| | Did not work/household duties | 913 (47.1) | 465 (50.9) | | 749 (82.0) | |

*(Continued)*

**Table 1.** (Continued)

| | | N (%) | Breastfeeding within first hour N (%)[b] | P-value[c] | Age-appropriate feeding practice N (%)[b] | P-value[c] |
|---|---|---|---|---|---|---|
| | Agricultural work (paid and unpaid) | 803 (41.4) | 481 (59.9) | | 719 (89.4) | |
| | Non-agricultural work (paid) | 221 (11.4) | 126 (57.0) | | 197 (89.1) | |
| *Highest education level*[d] | | | | **0.074** | | 0.901 |
| | No education | 555 (28.6) | 275 (49.5) | | 473 (85.2) | |
| | Primary | 380 (19.7) | 220 (57.7) | | 324 (85.3) | |
| | Secondary | 707 (36.5) | 415 (58.7)) | | 612 (86.6) | |
| | Higher education (tertiary +) | 295 (15.2) | 163 (55.3) | | 255 (86.4) | |
| *Antenatal and perinatal service delivery* | | | | | | |
| *Number of antenatal visits* | | | | 0.189 | | 0.159 |
| | 0 visits | 69 (3.6) | 36 (52.2) | | 64 (92.8) | |
| | 1–3 visits | 483 (24.9) | 248 (51.3) | | 406 (84.1) | |
| | 4 visits | 580 (30.0) | 343 (59.0) | | 513 (88.4) | |
| | 5–7 visits | 634 (32.7) | 361 (56.9) | | 540 (85.2) | |
| | 8+ visits | 172 (8.9) | 85 (49.7) | | 141 (82.0) | |
| *Type of delivery* | | | | **<0.001** | | 0.854 |
| | Vaginal | 1742 (89.9) | 1029 (59.0) | | 1496 (85.9) | |
| | Caesarean | 195 (10.1) | 44 (22.6) | | 169 (86.2) | |
| *Place of delivery* | | | | **<0.001** | | **0.084** |
| | Home | 666 (34.4) | 316 (47.5) | | 567 (85.1) | |
| | Health facility | 1178 (60.8) | 709 (60.2) | | 1025 (87.0) | |
| | Other | 94 (4.8) | 47 (50.5) | | 73 (77.7) | |
| *Child put on mother's bare skin after birth*[e] | | | | **<0.001** | | 0.521 |
| | No | 695 (35.9) | 281(40.4) | | 592 (85.2) | |
| | Yes | 1222 (63.0) | 789 (64.6) | | 1054 (86.3) | |
| | Don't know[g] | 21 (1.1) | 22 (N/A) | | | |
| *Breastfeeding counselling from any healthcare provider during first 2 days* | | | | **0.002** | | 0.339 |
| | No | 882 (45.5) | 445 (50.4) | | 751 (85.1) | |
| | Yes | 1049 (54.1) | 627 (59.8) | | 912 (86.9) | |
| | Don't know[g] | 6(0.3) | | | | |
| *Observation of breastfeeding by any healthcare provider during first 2 days* | | | | **<0.001** | | 0.344 |

(*Continued*)

**Table 1.** (Continued)

| | | N (%) | Breastfeeding within first hour N (%)[b] | P-value[c] | Age-appropriate feeding practice N (%)[b] | P-value[c] |
|---|---|---|---|---|---|---|
| | No | 959 (49.5) | 477 (49.8) | | 816 (85.1) | |
| | Yes | 974 (50.3) | 596 (61.3) | | 847 (87.0) | |
| | Don't know[g] | 5 (0.2) | | | | |
| *Household level factors* | | | | | | |
| *Wealth quintile* | | | | 0.133 | | 0.218 |
| | Poorest | 405 (20.9) | 252 (62.2) | | 364 (89.7) | |
| | Poorer | 408 (21.0) | 221 (54.2) | | 352 (86.3) | |
| | Middle | 444 (22.9) | 230 (52.3) | | 379 (85.4) | |
| | Richer | 398 (20.6) | 227 (57.0) | | 334 (83.9) | |
| | Richest | 283 (14.6) | 144 (51.4) | | 237 (83.7) | |
| *Husband's education level[f]* | | | | **0.025** | | 0.794 |
| | No education | 255 (13.1) | 131 (51.4) | | 220 (86.6) | |
| | Primary | 435 (22.5) | 218 (50.2) | | 368 (84.6) | |
| | Secondary | 863 (44.6) | 516 (59.9) | | 748 (86.6) | |
| | Higher education (tertiary +) | 377 (19.5) | 209 (55.9) | | 322 (85.2) | |
| | Don't know[g] | 2 (0.1) | | | | |
| *Ethnicity* | | | | **0.001** | | **<0.001** |
| | Brahman/Chhetri | 524 (27.0) | 322 (61.6) | | 464 (88.7) | |
| | Other Terai Castes | 387 (20.0) | 178 (46.1) | | 316 (81.4) | |
| | Dalits | 268 (13.8) | 150 (56.4) | | 223 (83.2) | |
| | Newar | 61 (3.1) | 20 (32.8) | | 58 (95.1) | |
| | Janajati | 546 (28.2) | 319 (59.0) | | 487 (89.2) | |
| | Muslim and other | 151 (7.8) | 83 (54.6) | | 117 (77.0) | |
| *Food Security* | | | | 0.911 | | 0.493 |
| | Secure | 791 (40.8) | 435 (55.4) | | 674 (85.2) | |
| | Not secure | 1147 (59.2) | 637 (55.7) | | 991 (86.4) | |
| *Community level factors* | | | | | | |
| *Ecological zone[d]* | | | | **0.031** | | **0.007** |
| | Terai | 1065 (55.0) | 555 (52.4) | | 887 (83.3) | |
| | Mountain | 127 (6.5) | 78 (61.9) | | 112 (88.2) | |

*(Continued)*

**Table 1.** (Continued)

| | | N (%) | Breastfeeding within first hour N (%)[b] | P-value[c] | Age-appropriate feeding practice N (%)[b] | P-value[c] |
|---|---|---|---|---|---|---|
| | Hill | 746 (38.5) | 440 (59.2) | | 665 (89.3) | |
| *Province* | | | | <0.001 | | <0.001 |
| | Province 1 | 335 (17.3) | 173 (51.6) | | 285 (84.8) | |
| | Province 2 | 501 (25.8) | 227 (45.6) | | 394 (78.6) | |
| | Province 3 | 305 (15.7) | 175 (57.9) | | 266 (87.2) | |
| | Province 4 | 162 (8.4) | 89 (54.9) | | 147 (90.7) | |
| | Province 5 | 354 (18.3) | 213 (60.2) | | 318 (89.6) | |
| | Province 6 | 118 (6.1) | 81 (68.6) | | 104 (88.1) | |
| | Province 7 | 162 (8.4) | 115 (71.9) | | 151 (93.2) | |
| *Type of place of residence* | | | | 0.111 | | 0.852 |
| | Urban | 1042 (53.8) | 600 (57.9) | | 894 (85.7) | |
| | Rural | 895 (46.2) | 473 (52.9) | | 771 (86.0) | |

Notes:

Bold text indicates P-value <0.1 and included in subsequent regression models.

N/A Not applicable.

[a] Column percentage.

[b] Row percentage.

[c] P-value from complex sample chi-squared tests.

[d] One missing value excluded from analysis.

[e] Two missing values excluded from analysis.

[f] Five missing value excluded from analysis.

[g] 'Don't know' values excluded from further analysis.

breastfeeding within two days post-delivery by a healthcare provider (p<0.001), ethnicity (p = 0.001), ecological zone (p = 0.031) and province (p<0.001). Since counselling regarding breastfeeding within two days of delivery and observation of breastfeeding within two days of delivery were highly associated with each other (p<0.0005), only one of these variables, observation of breastfeeding within two days of delivery, was included in the multivariable model.

After adjustment, infants put on their mother's bare skin immediately after birth had 2.10 times the odds of being breastfed within one hour of delivery compared with those who were not (95% CI: 1.63,2.72) (Table 2). The observation of breastfeeding by a healthcare provider within two days of delivery was also associated with increased odds of having been breastfed within the first hour (aOR = 1.58, 95% CI:1.20,2.08). Those infants delivered vaginally had markedly greater odds of being breastfed within the first hour, compared with those delivered via caesarean section (aOR: 4.76, 95% CI: 2.96,7.65).

Overall, there was a high level of adherence to age-appropriate feeding practice according to the IYCF guidelines (85.9%) (Table 1). However, there were differences according to the age of the child. Exclusive breastfeeding of infants aged 0–1 month was 79.4% but was only 40.8%

**Table 2. Unadjusted and adjusted odds ratios for factors influencing breastfeeding within one hour post-delivery and age-appropriate infant and young child feeding (IYCF) practice (n = 1938).**

| | Breastfeeding within first hour Unadjusted OR (95% CI) | Breastfeeding within first hour Adjusted OR (95% CI) | Age-appropriate feeding practice Unadjusted OR (95% CI) | Age-appropriate feeding practice Adjusted OR (95% CI) |
|---|---|---|---|---|
| **Sex of child** | | | | |
| Male | 1.00 Ref | 1.00 Ref | 1.00 Ref | 1.00 Ref |
| Female | 1.160 (0.95–1.42) | 1.24 (1.00–1.54) | 1.00 (0.74–1.37) | 0.99 (0.679–1.435) |
| **Age of child (months)** | | | | |
| 0–1 month | N/A | | 1.00 Ref | 1.00 Ref |
| 2–3 months | | | 0.67 (0.36–1.26) | 0.68 (0.35–1.32) |
| 4–5 months | | | **0.18 (0.01–0.33) \*\*\*** | **0.16 (0.08–0.30) \*\*\*** |
| 6–8 months | | | 1.27 (0.72–2.26) | 1.31 (0.72–2.39) |
| 9–11 months | | | **4.83 (2.12–10.97) \*\*\*** | **5.09 (2.25–11. 50) \*\*\*** |
| 12–17 months | | | **5.57 (2.74–11.32) \*\*\*** | **5.60 (2.73–11.47) \*\*\*** |
| 18–23 months | | | **2.39 (1.31–4.39) \*\*** | **2.46 (1.31–4.64) \*\*** |
| **Age of mother (years)[a]** | | | | |
| 15–19 | 0.99 (0.74–1.31) | | **0.65 (0.45–0.92) \*** | 0.85(0.52–1.40) |
| 20–29 | 1.00 Ref | | 1.00 Ref | 1.00 Ref |
| 30–39 | 1.02 (0.75–1.40) | | 0.96 (0.64–1.46) | 0.90 (0.56–1.43) |
| 40–49 | 0.57 (0.26–1.26) | | **10.60 (1.41–79.56) \*** | **7.35 (1.14–47.29) \*** |
| **Mother's highest education level** | | | | |
| No education | 1.00 Ref | 1.00 Ref | 1.00 Ref | |
| Primary | 1.39 (0.99–1.96) | 1.25 (0.86–1.82) | 1.00 (0.65–1.56) | |
| Secondary | **1.45 (1.05–1.99) \*** | 1.10 (0.80–1.52) | 1.13 (0.78–1.61) | |
| Higher education | 1.25 (0.82–1.90) | 1.00 (0.62–1.60) | 1.11 (0.66–1.88) | |
| **Mother's occupation** | | | | |
| Did not work/ household duties | 1.00 Ref | 1.00 Ref | 1.00 Ref | 1.00 Ref |
| Agricultural work (paid and unpaid) | **1.44 (1.15–1.80) \*\*** | 1.21 (0.94–1.57) | **1.86 (1.35–2.56) \*\*\*** | 1.27 (0.83–1.95) |
| Non-agricultural work (paid) | 1.29 (0.87–1.90) | 1.26 (0.83–1.91) | **1.80 (1.02–3.18) \*** | 1.15 (0.62–2.13) |
| **Husband's highest education level** | | | | |
| No education | 1.00 Ref | 1.00 Ref | 1.00 Ref | |
| Primary | 0.95 (0.68–1.32) | 0.79 (0.55–1.13) | 0.85 (0.53–1.38) | |
| Secondary | 1.41 (0.99–2.00) | 1.16 (0.82–1.64) | 1.01 (0.65–1.58) | |
| Higher education | 1.17 (0.76–1.81) | 0.93 (0.60–1.46) | 0.91 (0.55–1.49) | |
| **Type of delivery** | | | | |
| Caesarean | 1.00 Ref | 1.00 Ref | 1.00 Ref | |
| Vaginal | **4.95 (3.31–7.43) \*\*\*** | **4.76 (2.96–7.65) \*\*\*** | 0.96 (0.59–1.55) | |
| **Place of delivery** | | | | |
| Home | 1.00 Ref | 1.00 Ref | 1.00 Ref | 1.00 Ref |
| Health facility | **1.67 (1.29–2.15)\*\*\*** | 1.29 (0.94–1.78) | 1.17 (0.85–1.62) | 1.10 (0.75–1.62) |
| Other | 1.13 (0.67–1.90) | 0.95 (0.55–1.63) | 0.60 (0.31–1.15) | 0.66 (0.29–1.53) |
| **Child put on mother's bare skin immediately after birth[b]** | | | | |
| No | 1.00 Ref | 1.00 Ref | 1.00 Ref | |

(*Continued*)

**Table 2.** (Continued)

| | Breastfeeding within first hour Unadjusted OR (95% CI) | Breastfeeding within first hour Adjusted OR (95% CI) | Age-appropriate feeding practice Unadjusted OR (95% CI) | Age-appropriate feeding practice Adjusted OR (95% CI) |
|---|---|---|---|---|
| Yes | **2.69 (2.16–3.33)** *** | **2.10 (1.63–2.72)** *** | 1.10 (0.83–1.46) | |
| *Breastfeeding counselling from any healthcare provider during first 2 days* | | | | |
| No | 1.00 Ref | | 1.00 Ref | |
| Yes | **1.46 (1.17–1.83)** *** | | 1.16 (0.86–1.57) | |
| *Observation of breastfeeding by any healthcare provider during first 2 days* | | | | |
| No | 1.00 Ref | 1.00 Ref | 1.00 Ref | |
| Yes | **1.60 (1.28–2.00)** *** | **1.58 (1.20–2.08)** ** | 1.16 (0.85–1.58) | |
| *Ethnicity* | | | | |
| Brahman/Chhetri | 1.00 Ref | 1.00 Ref | 1.00 Ref | 1.00 Ref |
| Other Terai Castes | **0.53 (0.37–0.78)** ** | 1.18 (0.70–1.97) | **0.56 (0.37–0.87)** ** | 0.75 (0.44–1.29) |
| Dalits | 0.80 (0.55–1.16) | 1.16 (0.76–1.76) | 0.63 (0.38–1.04) | 0.60 (0.35–1.02) |
| Newar | **0.31 (0.13–0.75)** * | 0.46 (0.19–1.10) | 2.49 (0.58–10.73) | 2.39 (0.61–9.42) |
| Janajati | 0.88 (0.65–1.20) | 1.21 (0.87–1.67) | 1.06 (0.67–1.66) | 1.07 (0.66–1.74) |
| Muslim and other | 0.76 (0.50–1.14) | 1.55 (0.89–2.68) | **0.44 (0.27–0.71)** *** | 0.54 (0.29–1.02) |
| *Ecological zone*[a] | | | | |
| Terai | **0.76 (0.58–0.98)** * | 0.98 (0.68–1.42) | **0.60 (0.43–0.85)** ** | 0.86 (0.56–1.30) |
| Mountain | 1.12 (0.75–1.67) | 1.28 (0.76–2.14) | 0.92 (0.47–1.80) | 0.82 (0.42–1.60) |
| Hill | 1.00 Ref | 1.00 Ref | 1.00 Ref | 1.00 Ref |
| *Province* | | | | |
| Province 1 | 0.79 (0.50–1.24) | 0.70 (0.42–1.14) | 0.83 (0.45–1.52) | 0.96 (0.53–1.76) |
| Province 2 | **0.62 (0.41–0.93)** * | **0.51 (0.29–0.87)** * | **0.54 (0.31–0.96)** * | 0.98 (0.50–1.91) |
| Province 3 | 1.00 Ref | 1.00 Ref | 1.00 Ref | 1.00 Ref |
| Province 4 | 0.90 (0.59–1.38) | 0.79 (0.51–1.22) | 1.44 (0.71–2.93) | 1.65 (0.84–3.25) |
| Province 5 | 1.12 (0.74–1.68) | 0.85 (0.54–1.32) | 1.27 (0.69–2.35) | **1.93 (1.02–3.67)** * |
| Province 6 | **1.61 (1.01–2.56)** * | **1**.40 (0.88–2.24) | 1.14 (0.58–2.22) | 1.69 (0.81–3.51) |
| Province 7 | **1.85 (1.23–2.77)** ** | 1.31 (0.83–2.07) | **2.04 (1.07–3.89)** * | **3.11 (1.54–6.30)** ** |
| **Factors included in model** | | Sex of child, mother's highest education level, mother's occupation, husband's highest education level, type of delivery, place of delivery, child put on mother's bare skin immediately after birth, observation of breastfeeding by healthcare provider during first 2 days, ethnicity, ecological zone, province | | Sex of child, age of child (months), age of mother (years), mother's occupation, place of delivery, ethnicity, ecological zone, province |

Notes:

Bold text indicates statistically significant result.

*p<0.05

** p<0.01

*** p<0.001.

N/A Not applicable.

[a] 1 missing value that was excluded from analysis.

[b] 2 missing values that were excluded from analysis.

in infants aged 4–5 months. The rate of continued breastfeeding with complementary feeding ranged from 83.4% (6–8 months) to 95.6% (12–17 months).

The variables identified by Chi-square tests to be significantly associated with age-appropriate feeding practice included: the age of the child (p<0.0005), the mother's age (p = 0.016), the mother's occupation (p = 0.001), ethnicity (p<0.001), ecological zone (p = 0.007) and province (p<0.001).

After adjustment, the current age of the child significantly influenced the odds of age-appropriate feeding practice (Table 2). Mothers with infants aged 4–5 months had significantly lower odds of exclusive breastfeeding than those with infants aged 0–1 month (aOR = 0.16, 95% CI: 0.08,0.30). No variables related to antenatal or perinatal service delivery (number of antenatal visits, delivery in a health facility, vaginal delivery, infant-mother skin-to-skin contact post-delivery, the observation of breastfeeding within the first two days after delivery) were significantly associated with age-appropriate feeding practice.

The results of the analyses performed on the two age subgroups, 0–5 months (exclusive breastfeeding) and 6–23 months (continued breastfeeding with complementary foods), are presented in S1 Table (Chi-square analyses) and S2 Table (multivariable logistic regression analyses). The results of the main analysis (aggregated age-group) are included for comparison. In summary, the results for exclusive breastfeeding (0–5 months) after adjustment were almost identical to the main analysis, with mothers with infants aged 4–5 months having significantly lower odds of exclusive breastfeeding than those with infants aged 0–1 month (aOR = 0.16, 95% CI: 0.09,0.30). No variables related to antenatal or perinatal service delivery were significantly associated with exclusive breastfeeding after adjustment. In the 6–23 month age-group the variables associated with continued breastfeeding with complementary foods were also very similar to those of the main analysis after adjustment, noting minor differences in demographic associations. Children aged 6–8 months and 18–23 months had significantly lower odds of receiving both breastmilk and complementary foods (aOR = 0.20, 95% CI: 0.09,0.45 and aOR = 0.41, 95% CI: 0.20,0.85 respectively) compared to those aged 12–17 months (95.6% receiving both breastmilk and complementary foods). No variables related to antenatal or perinatal service delivery were significantly associated with continued breastfeeding with complementary foods after adjustment.

## Discussion

The aim of this study was to determine whether antenatal and perinatal service delivery are associated with early breastfeeding initiation and age-appropriate feeding practice amongst Nepali mother-infant dyads. Our analysis showed that several perinatal services were significantly associated with early breastfeeding initiation after adjustment for confounders: vaginal delivery, infant-mother skin-to-skin contact post-delivery, and the observation of breastfeeding within the first two days after delivery. By contrast, while some demographic factors (child's age, mother's age and province) were significantly associated with age-appropriate feeding practice, none of the antenatal or perinatal service delivery factors examined (number of antenatal visits, delivery in a health facility, vaginal delivery, infant-mother skin-to-skin contact post-delivery, the observation of breastfeeding within the first two days after delivery) were significantly associated after adjustment for confounders.

### Breastfeeding within the first hour after delivery

Although vaginal delivery has been associated with early initiation of breastfeeding in Nepal previously [17, 26], the association of infant-mother skin-to-skin contact post-delivery with breastfeeding within one hour of delivery has not previously been demonstrated in this

population. The low rate of infant-mother skin-to-skin contact post-delivery after caesarean delivery observed in our study (37.4% compared to 63.0% overall) may help explain the lower rate of breastfeeding within the first hour after caesarean section compared to vaginal delivery. Although the temporality of the relationship between observation of breastfeeding by a health-care provider in the first two days after delivery and breastfeeding within the first hour cannot be discerned here, it is plausible that breastfeeding within one hour of delivery may be facilitated when health professionals are available for observation and troubleshooting post-delivery.

Unlike previous research that demonstrated improved odds of early initiation of breastfeeding with health facility delivery [17], higher maternal education [12, 19] and large infant size at birth [12], these were not significant explanatory variables in the present study after adjustment. The observation of breastfeeding by a healthcare provider within the first two days post-delivery and infant-mother skin-to-skin contact post-delivery were more common for births within a health facility than not in a health facility (71.5% vs. 17.4% and 76.6% vs. 43.9% respectively); this confounding may explain why health facility delivery was not significant in this analysis.

The low rate (55.6%) of Nepali newborns breastfed in the first hour of life could be related to gaps in the provision of infant-mother skin-to-skin contact post-delivery and breastfeeding observation by healthcare professionals [10, 27]. The relationship between infant-mother skin-to-skin contact post-delivery and breastfeeding within one hour of delivery has been observed in other South Asian countries [28, 29] and supports the WHO recommendation for infant-mother skin-to-skin contact post-delivery for the initiation and establishment of breastfeeding [21]. Further prospective studies and service improvement activities focussing on the implementation of breastfeeding policies in Nepal are needed to improve the rate of newborns breastfed in the first hour of life, including the WHO / UNICEF Baby Friendly Hospital Initiative [10] and the training of skilled birth attendants. A recent study has shown skilled birth attendants in Nepal are not receiving either effective training or sufficient practice to stay clinically competent and knowledgeable [27].

## Age-appropriate feeding practice

The lower odds of exclusive breastfeeding of infants aged 4–5 months compared to those aged 0-1 month may reflect the challenges of exclusive breastfeeding as infants approach 6 months of age. Less than half of infants (40.8%) were exclusively breastfed at 4–5 months; other studies in Nepal have also demonstrated decreased exclusive breastfeeding approaching 6 months post-birth [19, 30]. Multiple psychological, social, physiological and financial barriers to exclusive breastfeeding may exist. Lack of time for breastfeeding due to work and mothers' perception of inadequate milk supply have been identified as the main reasons for starting complementary food early in Nepal [30, 31]. Physiologically, 24-hour milk intake does not increase in many infants between 3 and 6 months and may lead to a misconception of inadequate milk supply [32]. The lack of association between antenatal or perinatal service delivery and exclusive breastfeeding to 6 months of age in our study suggests a need to include information about exclusive breastfeeding in these services and to provide additional postnatal support. Another study found very few mothers received any information on breastfeeding during antenatal visits in Nepal [30].

From 6–23 months of age the continuation of breastfeeding plus "complementary foods" ranged from 83–96%, indicating that from 6 months of age onwards age-appropriate feeding is mostly being achieved but with room for improvement in the timely introduction of complementary foods at age 6 months, and continued breastfeeding until two years of age.

## Strengths and limitations

The NDHS is a large-scale population-based survey that attempts to be nationally representative by means of multi-stage sampling. However, some limitations warrant consideration. Due to the cross-sectional design, this study cannot determine whether relationships between explanatory and outcome variables are causal. Other limitations relate to the presence of measurement bias. The variables "observation of breastfeeding by any healthcare provider within the first two days post-delivery" and "breastfeeding counselling by any healthcare provider within the first two days post-delivery" were measured in a dichotomous "yes/no" format and the exact timings of observation or counselling were not recorded. However, in our analysis we assumed that breastfeeding observation and counselling preceded the event, "breastfeeding within the first hour". Also, the size of the infant at birth was self-reported by the mother as small / very small, average, or large, which is inaccurate and subjective. Although this study attempted to minimise potential recall bias by including only last-born children born in the 24 months preceding the survey, recall bias may still exist. Unmeasured confounders may also have influenced the associations observed. This study was focussed on breastfeeding outcomes and did not evaluate the nutritional quality or quantity of complementary foods. Future longitudinal observational studies may offer opportunities to demonstrate cause-effect relationships between antenatal and perinatal healthcare delivery and breastfeeding outcomes.

## Conclusions for practice

This study highlights a positive association between perinatal service delivery and initiation of breastfeeding within the first hour after delivery. Further promotion of infant-mother skin-to-skin contact post-delivery (including after caesarean delivery) and early breastfeeding observation and counselling, may increase the rates of breastfeeding within one hour of delivery from just over 50% and, downstream, improve infant health. We also identified a large gap in exclusive breastfeeding of infants aged 4–5 months of age which was not associated with the number of antenatal visits or perinatal service delivery. The promotion of exclusive breastfeeding in these services, and additional postnatal support for breastfeeding mothers with infants approaching 4–5 months of age, should be considered to address the low rate (40.8%) of exclusive breastfeeding observed in this group.

These service improvements could be achieved by strengthening the implementation, monitoring and supervision of the 10 steps of the WHO / UNICEF Baby Friendly Hospitals Initiative (BFHI) [10]. Similarly, the strengthening of training and sufficient practice for skilled birth attendants [27] in breastfeeding support will be important to improve breastfeeding outcomes.

## Supporting information

**S1 Table. Distribution of age-appropriate infant and young child (IYCF) feeding practice according to infant age group.**
(DOCX)

**S2 Table. Adjusted odds ratios for age-appropriate infant and young child feeding (IYCF) feeding practice according to infant age group.**
(DOCX)

## Acknowledgments

We thank the Australian Government for providing BH with a short-term mobility scholarship from the Endeavour Leadership Program to undertake research training in Nepal. We are

also grateful to the DHS program, funded by USAID, for conducting the survey in Nepal and making their data available to us for research purposes.

## Author Contributions

**Conceptualization:** Breanna Louise Hollow, Hilary Jane Wallace.

**Data curation:** Breanna Louise Hollow, Prakash Dev Pant.

**Formal analysis:** Breanna Louise Hollow, Max K. Bulsara, Hilary Jane Wallace.

**Methodology:** Max K. Bulsara, Prakash Dev Pant, Hilary Jane Wallace.

**Project administration:** Hilary Jane Wallace.

**Resources:** Max K. Bulsara, Prakash Dev Pant, Hilary Jane Wallace.

**Software:** Prakash Dev Pant.

**Supervision:** Max K. Bulsara, Prakash Dev Pant, Hilary Jane Wallace.

**Writing – original draft:** Breanna Louise Hollow, Hilary Jane Wallace.

**Writing – review & editing:** Breanna Louise Hollow, Max K. Bulsara, Prakash Dev Pant, Hilary Jane Wallace.

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
