## [Decision Letter · Decision Letter 0]

24 Nov 2022

PGPH-D-22-01442

Antenatal and perinatal service delivery associations with breastfeeding outcomes in Nepal: Analysis of the 2016 Nepal Demographic and Health Survey

Dear Dr. Wallace,

Thank you for submitting your manuscript to PLOS Global Public Health. After careful consideration, we feel that it has merit but does not fully meet PLOS Global Public Health’s publication criteria as it currently stands. Therefore, we invite you to submit a revised version of the manuscript that addresses the points raised during the review process.

We invite you to submit a revised version of your manuscript, taking into consideration the comments received from the three reviewers. In particular, we agree with the comment of reviewer #1 regarding the need to clarify the indicator related to observed feeding and counselling, and highlight this as a limitation. Please provide a rationale for why you decided to treat antenatal care as a dichotomous variable. Finally, please consider disaggregating the variable, age-appropriate breastfeeding practice, by age category.

We look forward to receiving your revised manuscript.

Kind regards,

Melissa Morgan Medvedev, M.D., Ph.D.

Academic Editor

Journal Requirements:

1. Please the 'Competing Interests' statement and state "The authors have declared that no competing interests exist".

a. State what role the funders took in the study. If the funders had no role in your study, please state: “The funders had no role in study design, data collection and analysis, decision to publish, or preparation of the manuscript.”

3. Please update the Funding Information in the system and ensure that it matches with the Financial Disclosure Statement.

4. Please provide separate figure files in .tif or .eps format.

5. We noticed that you used “data not shown”/"unpublished data" in the manuscript. We do not allow these references, as the PLOS data access policy requires that all data be either published with the manuscript or made available in a publicly accessible database. Please amend the supplementary material to include the referenced data or remove the references. 

Additional Editor Comments (if provided):

We invite you to submit a revised version of your manuscript, taking into consideration the comments received from the three reviewers. In particular, we agree with the comment of reviewer #1 regarding the need to clarify the indicator related to observed feeding and counselling, and highlight this as a limitation. Please provide a rationale for why you decided to treat antenatal care as a dichotomous variable. Finally, please consider disaggregating the variable, age-appropriate breastfeeding practice, by age category.

Reviewers' comments:

Reviewer's Responses to Questions

**Comments to the Author**

1. Does this manuscript meet PLOS Global Public Health’s publication criteria? Is the manuscript technically sound, and do the data support the conclusions? The manuscript must describe methodologically and ethically rigorous research with conclusions that are appropriately drawn based on the data presented.

Reviewer #1: Yes

Reviewer #2: Yes

Reviewer #3: Yes

2. Has the statistical analysis been performed appropriately and rigorously?

Reviewer #1: Yes

Reviewer #2: Yes

Reviewer #3: Yes

3. Have the authors made all data underlying the findings in their manuscript fully available (please refer to the Data Availability Statement at the start of the manuscript PDF file)?

Reviewer #1: Yes

Reviewer #2: Yes

Reviewer #3: Yes

4. Is the manuscript presented in an intelligible fashion and written in standard English?

Reviewer #1: Yes

Reviewer #2: Yes

Reviewer #3: Yes

5. Review Comments to the Author

Reviewer #1: Overall:

This paper presents an analysis of an important area. Early initiation of exclusive breastfeeding and age-appropriate breastfeeding are well recognised as interventions to reduce infant and child morbidity and mortality. As such it deserves to be published and authors should be given a chance to resubmit.

Major comments

Overall the paper would benefit from some clarity in the indicators. In particular, the indicator related to observed feeding and counselling. This is often referred to in a confusing way, especially when in relation to skin to skin contact and early initiation of breastfeeding. The indicator is defined as being measured over 2 days compared to the first hour after birth. The association is therefore difficult to interpret unless we assume that the observation took place within the first hour after birth. Please do consider making clearer throughout the text and also highlighting it as a limitation.

I also find the order in which results are presented hard to follow. I think presenting skin to skin, then early initiation, then age appropriate feeding practices is a more logical order and would improve the flow of the paper.

Minor comments

Abstract – I don’t think the abstract concisely captures the main points of the paper. Some reworking would benefit the paper and make it more inviting to readers

Table 1 - Please define weights for size at birth

Line 201 – the term recent is confusing in this sentence

Lines 220-223 – please could you elaborate on the meaning of this a little more

Page 17 (sorry, line numbers disappeared) – I think the discussion on supply could benefit from further information. Are there physiological reasons for diminished supply or is it due to mixed feeding. Do other studies exist or is this a research gap in this context? Is it mothers perception of supply? Please define how this is determined.

Page 18 – 3rd paragraph – please pull apart and comment more clearly on what is needed in terms of early breastfeeding advice. For example, is it the responsibility of skilled birth attendants to promote immediate skin to skin and the support the mother within the first hour following birth with attachment and feeding? Also how can this be improved for women with non vaginal deliveries? Would promotion of skin to skin and early attachment among obstetricians performing caesareans help? Similarly, with those who do not have skilled attendance. Are there local or regional examples of good practice and impact that can be cited?

Reviewer #2: This is a well written artcile which meets all of the Journal's publication criteras like originality, not been previously done, not published elsewhere etc. It has also applied rigorous statistical methods for analysis and explains the process in the text. The sample size is big enough and criterias for selection of respondents have been explained well. The primary data it uses is a publicly shared one and it has been mentioned and source has been cited in the article.

Reviewer #3: This manuscript examines factors related to antenatal and perinatal service delivery and breastfeeding outcomes using the 2016 Nepal DHS. The paper is well written. The analysis is conducted using a fairly standard approach, with selection of factors for the multivariable model based on both theory and significance testing in unadjusted models and appropriately adjusts for survey weights and the cluster design of the survey. Findings from the analysis are generally aligned with expectations based on previous studies in Nepal and globally. I just have a few thoughts that I think could improve the paper.

Lines 165-167: The sentence beginning with “Where two explanatory variables” needs further explanation. It’s not clear exactly what this means but if whether the reason to select one variable is due to multicollinearity concerns, I would recommend looking at the VIF for each variable.

I have a couple of concerns about the “age-appropriate breastfeeding practice” outcome variable. I can see how on its surface it seems like it might be useful as a general reflection of the proportion of under-24 month olds that are being fed according to guidelines. However, the variable is very influenced by age, as illustrated in the proportion of infants/young children that adhere to it in Table 1: there is a steep rise from 6 months onward that corresponds to the inclusion of complementary foods in its makeup. For this reason, it seems to me that a presentation of this variable disaggregated by age category would be more useful, you may find that different factors are associated with it for different age groups.

I am curious why you elected to treat ANC receipt as a dichotomous variable.

Line 230.. and elsewhere isn’t it age appropriate “feeding” not “breastfeeding” given that complementary foods?

Table 2. I assume there are typos in the OR’s for age in months here for older age groups.

Table 2. Is there a reason to not include the unadjusted OR’s for the age-appropriate feeding practices here? (they are blank)

Given that significance testing was used to determine which variables were included in final models, when reflecting on these findings vs. previous studies I’d suggest also looking at the unadjusted coefficients.. because the relative strength of associations and other variables that are included in your models are influencing what gets selected.

6. PLOS authors have the option to publish the peer review history of their article (what does this mean?). If published, this will include your full peer review and any attached files.

**Do you want your identity to be public for this peer review?** For information about this choice, including consent withdrawal, please see our Privacy Policy.

Reviewer #1: No

Reviewer #2: **Yes: **Mukta Singh Bhandari

Reviewer #3: No

---

## [Decision Letter · Decision Letter 1]

10 Feb 2023

PGPH-D-22-01442R1

Antenatal and perinatal service delivery associations with breastfeeding outcomes in Nepal: Analysis of the 2016 Nepal Demographic and Health Survey

Dear Dr. Wallace,

Thank you for submitting your manuscript to PLOS Global Public Health. After careful consideration, we feel that it has merit but does not fully meet PLOS Global Public Health’s publication criteria as it currently stands. Therefore, we invite you to submit a revised version of the manuscript that addresses the points raised during the review process.

We look forward to receiving your revised manuscript.

Kind regards,

Caroline Bull

Staff Editor

Journal Requirements:

Additional Editor Comments (if provided):

Reviewers' comments:

Reviewer's Responses to Questions

**Comments to the Author**

1. If the authors have adequately addressed your comments raised in a previous round of review and you feel that this manuscript is now acceptable for publication, you may indicate that here to bypass the “Comments to the Author” section, enter your conflict of interest statement in the “Confidential to Editor” section, and submit your "Accept" recommendation.

Reviewer #1: All comments have been addressed

Reviewer #3: (No Response)

2. Does this manuscript meet PLOS Global Public Health’s publication criteria? Is the manuscript technically sound, and do the data support the conclusions? The manuscript must describe methodologically and ethically rigorous research with conclusions that are appropriately drawn based on the data presented.

Reviewer #1: Yes

Reviewer #3: Yes

3. Has the statistical analysis been performed appropriately and rigorously?

Reviewer #1: Yes

Reviewer #3: Yes

4. Have the authors made all data underlying the findings in their manuscript fully available (please refer to the Data Availability Statement at the start of the manuscript PDF file)?

Reviewer #1: Yes

Reviewer #3: Yes

5. Is the manuscript presented in an intelligible fashion and written in standard English?

Reviewer #1: Yes

Reviewer #3: Yes

6. Review Comments to the Author

Reviewer #1: The authors have addressed most of my comments and concerns

Reviewer #3: I'm pasting my responses to the issues that i still have questions about below each of the previous responses beginning with OK (i wish we'd numbered them)

I am curious why you elected to treat ANC receipt as a dichotomous variable.

This was due to the original WHO recommendation of a minimum of 4 antenatal visits (0-3; 4+). We have now

disaggregated this variable into 0-3, 4, 5-7, 8+ to provide more granular information, and to reflect the new WHO

(2016) recommendation of 8+ antenatal visits (see lines 193-195 and Table 1). The new coding did not change

the relationship of number of antenatal visits with either outcome.

“Variables related to antenatal and perinatal service delivery were informed by the WHO breastfeeding

practice guidelines (21) and the WHO recommendations on antenatal care for a positive pregnancy

experience (23): number of antenatal visits (0-3 visits; 4 visits, 5-7 visits, 8+visits)…”

OK- at the time of this survey (2016) Nepal was recommending (and recording) 4 visits on ANC cards, this is changing. I do think there is value in assessing a finer stratification for those with <4 visits rather than lumping 0-3 together.. as 0 visits suggests non use of health services completely which is very different than seeking some care.. and 4 was “complete” adherence to recommendations at the time.

Line 230.. and elsewhere isn’t it age appropriate “feeding” not “breastfeeding” given that complementary foods?

We are focussed on the breastfeeding component and defined age-appropriate IYCF breastfeeding practice from

6-23 months as continued breastfeeding with complementary foods. We did not evaluate the quality or quantity

of complementary foods in this study, so did not wish to refer to age-appropriate ‘feeding’.

I find this confusing.. if complementary foods are included in the indicator, it’s more than just breastfeeding.

Table 2. I assume there are typos in the OR’s for age in months here for older age groups.

No – these are correct. We think the reviewer is referring to Age of mother (40-49 years). In this age-group the

numbers are very small and hence the 95% confidence interval of the odds ratio is very wide.

OK What I am looking at are the unadjusted OR’s and the corresponding 95% CI’s for age groups 12-17 months and 18-23 months. The 95% CI’s do not include the OR coefficients within their bounds.

Table 2. Is there a reason to not include the unadjusted OR’s for the age-appropriate feeding practices here?

(they are blank)

Not sure what the reviewer means here – have carefully checked Table 2 as submitted. Both unadjusted and

adjusted ORs are blank for variables that were not included in the models.

OK What I am suggesting is rather than leaving blank space for something that was actually tested, that you include the unadjusted OR’s in that space. This also would enable comparison vs.previous studies (as mentioned below).

7. PLOS authors have the option to publish the peer review history of their article (what does this mean?). If published, this will include your full peer review and any attached files.

**Do you want your identity to be public for this peer review?** For information about this choice, including consent withdrawal, please see our Privacy Policy.

Reviewer #1: No

Reviewer #3: No

---

## [Decision Letter · Decision Letter 2]

22 Mar 2023

Antenatal and perinatal service delivery associations with breastfeeding outcomes in Nepal: Analysis of the 2016 Nepal Demographic and Health Survey

PGPH-D-22-01442R2

Dear Authors,

We are pleased to inform you that your manuscript 'Antenatal and perinatal service delivery associations with breastfeeding outcomes in Nepal: Analysis of the 2016 Nepal Demographic and Health Survey' has been provisionally accepted for publication in PLOS Global Public Health.

Best regards,

Shela Hirani, PhD, IBCLC, RN

Academic Editor

Reviewer Comments (if any, and for reference):

Reviewer's Responses to Questions

**Comments to the Author**

1. If the authors have adequately addressed your comments raised in a previous round of review and you feel that this manuscript is now acceptable for publication, you may indicate that here to bypass the “Comments to the Author” section, enter your conflict of interest statement in the “Confidential to Editor” section, and submit your "Accept" recommendation.

Reviewer #1: All comments have been addressed

Reviewer #3: All comments have been addressed

2. Does this manuscript meet PLOS Global Public Health’s publication criteria? Is the manuscript technically sound, and do the data support the conclusions? The manuscript must describe methodologically and ethically rigorous research with conclusions that are appropriately drawn based on the data presented.

Reviewer #1: Yes

Reviewer #3: No

3. Has the statistical analysis been performed appropriately and rigorously?

Reviewer #1: Yes

Reviewer #3: Yes

4. Have the authors made all data underlying the findings in their manuscript fully available (please refer to the Data Availability Statement at the start of the manuscript PDF file)?

Reviewer #1: Yes

Reviewer #3: Yes

5. Is the manuscript presented in an intelligible fashion and written in standard English?

Reviewer #1: Yes

Reviewer #3: Yes

6. Review Comments to the Author

Reviewer #1: No further comments

Reviewer #3: looks fine now thank you

7. PLOS authors have the option to publish the peer review history of their article (what does this mean?). If published, this will include your full peer review and any attached files.

**Do you want your identity to be public for this peer review?** For information about this choice, including consent withdrawal, please see our Privacy Policy.

Reviewer #1: No

Reviewer #3: No
